# Effects of Sevoflurane and Fullerenol C60 on the Heart and Lung in Lower-Extremity Ischemia–Reperfusion Injury in Streptozotocin-Induced Diabetes Mice

**DOI:** 10.3390/medicina60081232

**Published:** 2024-07-29

**Authors:** Ender Örnek, Metin Alkan, Selin Erel, Zeynep Yığman, Ali Doğan Dursun, Aslı Dağlı, Badegül Sarıkaya, Gülay Kip, Yücel Polat, Mustafa Arslan

**Affiliations:** 1Department of Anesthesiology and Reanimation, Faculty of Medicine, Gazi University, Ankara 06560, Turkey; enornek@gmail.com (E.Ö.); metoalkan@gmail.com (M.A.); selinerel@yahoo.com (S.E.); gulaykip@yahoo.com (G.K.); 2Department of Histology and Embryology, Faculty of Medicine, Gazi University, Ankara 06560, Turkey; zeynepyigman@gmail.com (Z.Y.); dagli.asli@hotmail.com (A.D.); 3Neuroscience and Neurotechnology Center of Excellence (NOROM), Gazi University, Ankara 06560, Turkey; 4Department of Physiology, Faculty of Medicine, Atılım University, Ankara 06830, Turkey; alidogandursun@gmail.com (A.D.D.); badegul.sarikaya@atilim.edu.tr (B.S.); 5Medical Laboratory Techniques Program, Department of Medical Services and Techniques, Vocational School of Health Services, Atılım University, Ankara 06830, Turkey; 6Tekirdağ Dr İsmail Fehmi Cumalıoğlu City Hospital, Department of Cardiovascular Surgery, Tekirdağ 59030, Turkey; dr.yucelpolat@hotmail.com; 7Life Sciences Application and Research Center, Gazi University, Ankara 06560, Turkey; 8Laboratory Animal Breeding and Experimental Research Center (GUDAM), Gazi University, Ankara 06560, Turkey

**Keywords:** Fullerenol C60, diabetes, lower-limb ischemia, streptozotocin

## Abstract

*Background and Objectives*: Lower-extremity ischemia–reperfusion injury can induce distant organ ischemia, and patients with diabetes are particularly susceptible to ischemia–reperfusion injury. Sevoflurane, a widely used halogenated inhalation anesthetic, and fullerenol C60, a potent antioxidant, were investigated for their effects on heart and lung tissues in lower-extremity ischemia–reperfusion injury in streptozotocin (STZ)-induced diabetic mice. *Materials and Methods*: A total of 41 mice were divided into six groups: control (*n* = 6), diabetes–control (*n* = 7), diabetes–ischemia (*n* = 7), diabetes–ischemia–fullerenol C60 (*n* = 7), diabetes–ischemia–sevoflurane (*n* = 7), and diabetes–ischemia–fullerenol C60–sevoflurane (*n* = 7). Diabetes was induced in mice using a single intraperitoneal dose of 55 mg/kg STZ in all groups except for the control group. Mice in the control and diabetes–control groups underwent midline laparotomy and were sacrificed after 120 min. The DIR group underwent 120 min of lower-extremity ischemia followed by 120 min of reperfusion. In the DIR-F group, mice received 100 μg/kg fullerenol C60 intraperitoneally 30 min before IR. In the DIR-S group, sevoflurane and oxygen were administered during the IR procedure. In the DIR-FS group, fullerenol C60 and sevoflurane were administered. Biochemical and histological evaluations were performed on collected heart and lung tissues. *Results*: Histological examination of heart tissues showed significantly higher necrosis, polymorphonuclear leukocyte infiltration, edema, and total damage scores in the DIR group compared to controls. These effects were attenuated in fullerenol-treated groups. Lung tissue examination revealed more alveolar wall edema, hemorrhage, vascular congestion, polymorphonuclear leukocyte infiltration, and higher total damage scores in the DIR group compared to controls, with reduced injury parameters in the fullerenol-treated groups. Biochemical analyses indicated significantly higher total oxidative stress, oxidative stress index, and paraoxonase-1 levels in the DIR group compared to the control and diabetic groups. These levels were lower in the fullerenol-treated groups. *Conclusions*: Distant organ damage in the lung and heart tissues due to lower-extremity ischemia–reperfusion injury can be significantly reduced by fullerenol C60.

## 1. Introduction

Diabetes mellitus, a multifaceted chronic metabolic disorder, affects over 422 million people globally, emerging as a critical public health challenge [1]. Characterized by hyperglycemia resulting from insulin deficiency or dysfunction, DM is categorized into several types, each with distinct pathophysiological bases and associated risks. Complications arising from diabetes are severe, including both microvascular (retinopathy, neuropathy, nephropathy) and macrovascular conditions (cardiac disease, stroke, peripheral artery disease). These complications underscore the urgency in understanding and managing diabetes to mitigate its widespread prevalence and debilitating outcomes [2,3]. 

Ischemia, a major vascular complication in diabetic patients, involves tissue damage due to oxygen deprivation, while subsequent reperfusion can exacerbate injury through inflammatory responses [4]. This ischemia–reperfusion (IR) injury is pivotal in understanding the broader impacts of diabetes on organ systems, often leading to conditions such as distant organ injury and multiple organ dysfunction syndrome, significant contributors to mortality in critically ill patients [5]. Factors like xanthine oxidase, leukocytes, inflammatory mediators such as TNF-α and thromboxane B₂, alongside the complement system, play crucial roles in this context [6,7].

Transitioning into the realm of therapeutic interventions, sevoflurane, a preferred halogenated inhalation anesthetic since the 1970s, has demonstrated protective effects in the context of IR injury [8,9]. Studies on rat hearts have shown that sevoflurane helps maintain structural integrity during myocardial ischemia, prevents the opening of mitochondrial pores, reduces oxidative stress, suppresses mitophagy activation, and protects against ischemia–reperfusion injury through these mechanisms [10]. Sevoflurane prevents the shedding of the glycocalyx layer in the ischemic vessel wall by serving as a natural protector for endothelial adhesion molecules to prevent cell adhesion [9]. Sevoflurane has antioxidant and anti-inflammatory properties, thereby protecting tissues and organs from stress-induced damage [11]. 

Fullerenes are highly symmetrical cage-shaped molecules composed solely of carbon atoms that exhibit unique chemical and physical properties [12]. The areas of application of fullerenes are very broad, including toxicity to unwanted cells, antioxidant effects in ischemia–reperfusion injury, contrast enhancement agents in imaging techniques, cancer treatment, carriers in drug delivery systems, biosensor usage, and antiviral and antibacterial effects [13]. Fullerenes react with ROS molecules that exhibit increased oscillations during ischemia–reperfusion injury and play a role in the destruction of proteins, lipids, DNA, and other cellular components to prevent damage [14].

The lungs and heart are prone to ischemia. Lung ischemia–reperfusion injury can occur during lung transplantation, cardiopulmonary bypass, atherosclerosis, trauma, or pulmonary thromboembolism [15,16,17,18,19]. Pulmonary edema during ischemia and reperfusion impairs gas exchange and may lead to conditions such as acute lung injury or acute respiratory distress syndrome [20]. Acute myocardial infarction is the most common cause of death worldwide [21]. Cardiac ischemia causes cells to develop acidosis via anaerobic metabolism, leading to ventricular arrhythmia. Ischemic damage intensifies with reperfusion owing to increased reactive oxygen species production, calcium loading, and inflammatory activation [22,23]. 

Diabetes mellitus increases the susceptibility to ischemia–reperfusion injury. The mechanism of the exaggerated response to ischemia–reperfusion injury in diabetes has not been fully elucidated, but it is thought that hyperglycemia-induced oxidative stress plays a crucial role, leading to an imbalance between reactive oxygen species (ROS) production and the body’s antioxidant defense mechanisms [24]. Persistent oxidative stress, driven by both mitochondrial and non-mitochondrial sources, results in excessive superoxide production and subsequent cellular damage. Advanced glycation end products (AGEs) and their receptors (RAGEs) further exacerbate oxidative stress and contribute to vascular wall pathology by activating signaling pathways that induce inflammation and fibrosis. The chronic oxidative environment in diabetes also causes DNA damage and impairs endothelial function, disrupting normal vascular responses. Additionally, the inefficacy of ischemic conditioning strategies in diabetes is linked to impaired prosurvival signal transduction pathways, including alterations in adenosine signaling, the RISK pathway, and endothelial nitric oxide synthase (eNOS) activity [23]. 

These complex interactions culminate in heightened vulnerability to ischemic injury in diabetic individuals, underlining the need for targeted therapeutic approaches to mitigate unwanted effects. This study aims to illuminate how fullerenol C60, through its antioxidant properties, mitigates IR injury in the cardiac and pulmonary systems in diabetic mice models, focusing specifically on histopathological effects in these tissues.

## 2. Materials and Methods

### 2.1. Experimental Setup and Animals

The experiment was conducted in the animal laboratory of Gazi University Experimental Animals Application and Research Center (GUDAM) (G.Ü.E.T-23.066). This study used 41 adult male Swiss albino mice. Animals were housed under identical environmental conditions and kept in a temperature/humidity controlled room (20–21 °C, 45–55%) under a 12/12 h light/dark cycle. Food and water were available ad libitum. The mice were fasted for 2 h prior to anesthesia. Animals were divided into six random groups: control (C) (*n* = 6), diabetes–control (D) (*n* = 7), diabetes–ischemia (DIR) (*n* = 7), diabetes–ischemia–fullerenol C60 (DIR-F) (*n* = 7), diabetes–ischemia–sevoflurane (DIR-S) (*n* = 7), and diabetes–ischemia–fullerenol C60–sevoflurane (DIR-FS) (*n* = 7). 

### 2.2. Induction of Diabetes

A single dose of 55 mg/kg streptozotocin (Sigma Chemical, St. Louis, MO, USA) was administered intraperitoneally to induce diabetes [25,26]. Blood glucose levels were measured from tail vein samples 72 h post-injection using a glucometer (Standard GlucoNavii GGh, Korea). Mice with blood glucose levels ≥250 mg/dL were considered diabetic. Diabetic mice were monitored for four weeks after streptozotocin injection to confirm the establishment of chronic diabetes [27]. Insulin (1–3 U/day) was given to prevent weight loss and ketoacidosis over the next four weeks.

### 2.3. Anesthesia and Surgical Procedure

All mice were anesthetized with 5 mg/kg xylazine hydrochloride (Alfazyne 2%, Ege Vet.) and 50 mg/kg ketamine hydrochloride (Ketalar^®^, Pfizer PFE İlaçları, İstanbul, Turkey) intramuscularly. The procedures were performed under a heating lamp with the mice in the supine position. Following aseptic preparation, midline laparotomy was performed. In the ischemia–reperfusion groups, the intestines were retracted, and the infrarenal abdominal aorta was isolated and clamped to induce lower-extremity ischemia. After 120 min, the clamp was removed to allow for 120 min of reperfusion. Reperfusion was confirmed by the return of pulsation in the artery after the clamp was removed. During ischemia–reperfusion, saline was applied to the peritoneal cavity to minimize fluid and heat losses. Reperfusion was confirmed by the return of pulsation distal to the clamp.

### 2.4. Experimental Groups

Control group (C) (*n* = 6): Mice were anesthetized with 5 mg/kg xylazine hydrochloride and 50 mg/kg ketamine hydrochloride. Midline laparotomy was the sole surgical procedure without inducing ischemia–reperfusion. Hearts and lungs were harvested for analysis after sacrifice.

Diabetes–control group (D) (*n* = 7): Streptozin-induced diabetic mice were anesthetized and midline laparotomy was performed without ischemia and reperfusion. 

Diabetes–ischemia group (DIR) (*n* = 7): Streptozin-induced diabetic mice were anesthetized and midline laparotomy was performed. The infrarenal abdominal aorta was clamped for 120 min followed by unclamping for 120 min of reperfusion.

Diabetes–ischemia–fullerenol C60 group (DIR-F) (*n* = 7): 100 µg/kg fullerenol C60 was administered intraperitoneally 30 min before anesthesia to streptozotocin-induced diabetic mice. Then, ischemia–reperfusion was induced. 

Diabetes–ischemia–sevoflurane group (DIR-S) (*n* = 7): Diabetic mice were intramuscularly anesthetized with 5 mg/kg xylazine hydrochloride (Alfazyne 2%, Ege Vet.) and 50 mg/kg dose of ketamine hydrochloride (Ketalar^®^ vial; Parke Davis, PA, USA). After midline laparotomy, ischemia–reperfusion was induced by clamping the infrarenal abdominal aorta. During ischemia and reperfusion, the mice were kept in a transparent, sealed lantern with a hole for the gas inlet and outlet. A gas line device was set up, oxygen was administered through a flowmeter from a pressurized oxygen tank, and 2.3% sevoflurane was administered through a standard volatile agent vaporizer with a minimum alveolar concentration (MAC) of 1.

Diabetes–ischemia–fullerenol C60–sevoflurane group (DIR-FS) (*n* = 7): 100 µg/kg fullerenol C60 was administered intraperitoneally 30 min before anesthesia to streptozotocin-induced diabetic mice. Thirty minutes later, 5 mg/kg xylazine hydrochloride (Alfazyne 2%, Ege Vet.) and 50 mg/kg ketamine hydrochloride (Ketalar^®^ vial; Parke Davis, USA) were administered intramuscularly. After midline laparotomy, ischemia–reperfusion was induced by clamping the infrarenal abdominal aorta. During ischemia and reperfusion, the mice were kept in a transparent, sealed lantern with a hole for the gas inlet and outlet. In the meantime, a gas line device was set up, oxygen was administered through a flowmeter from a pressurized oxygen tank, and 2.3% sevoflurane was administered through a standard volatile agent vaporizer with a MAC of 1.

### 2.5. Tissue Collection and Analysis

At the end of the experiment, the mice were sacrificed by blood collection from the abdominal aorta, and hearts and lungs were harvested for histological and biochemical analyses. Histological samples were immersed in 10% neutral-buffered formalin for fixation.

### 2.6. Biochemical Analysis

The heart and lung samples were quickly frozen in liquid nitrogen and then kept at −70 °C in a deep freezer until they were analyzed for total antioxidant status (TAS), total oxidant status (TOS), oxidative stress index (OSI), and paraoxonase-1 (PON-1).

The procedures were carried out rapidly to prevent the tissues from thawing. Initially, 22 lancets (PLUSMED^®^, İstanbul, Türkiye) were used to cut portions (80–100 mg) from the tissues, which were weighed on a precision scale. The tissue was crushed in a porcelain bowl using liquid nitrogen, and the resulting powder was directly transferred to a homogenization tube (099C S3, Glas-Col). A solution of potassium chloride (KCl) with a concentration of 140 millimolar (mM) was added to each gram of tissue in order to obtain a dilution of 1/10 (weight/volume). In order to prevent temperature rise, the homogenization tube was positioned within a glass beaker filled with flake ice. The procedure was carried out using a Glas-Col (K5424^®^) homogenizer, operating at a speed of 50 rpm for a duration of 2 min, with the use of a pestle (099C S21G, Glas-Col). The homogenates were transferred to 1.5 mL Eppendorf tubes and subjected to centrifugation at a speed of 3000 revolutions per minute for a duration of 10 min (NF 048, NUVE). The supernatants obtained were transferred to new Eppendorf tubes [28].

Total oxidative status (TOS): TOS was measured using a RelAssay Diagnostics^®^ TOS kit (Turkey). According to the kit protocol, 500 µL of reagent 1 (measurement buffer) and 75 µL of the sample were mixed, and absorbance was measured at 530 nm using a NanoDrop^®^ ND-1000 spectrophotometer (A1). Next, 25 µL of reagent 2 (prochromogen solution) was added, and the mixture was incubated at 37 °C for 5 min. After incubation, absorbance was measured again at 530 nm (A2). The standard solution contained 10 µmol/L hydrogen peroxide (H_2_O_2_). Measurements were performed in triplicate and the changes in absorbance (∆Abs) were calculated. TOS levels were expressed in mmol H_2_O_2_ Eq/L [28].

Total antioxidant status (TAS): TAS was measured using a RelAssay Diagnostics^®^ TAS kit (Turkey). According to the kit protocol, 500 µL of reagent 1 (measurement buffer) and 30 µL of the sample were mixed, and absorbance was measured at 660 nm using a NanoDrop^®^ ND-1000 spectrophotometer (A1). Then, 75 µL of reagent 2 (ABTS solution) was added and the mixture was incubated at 37 °C for 5 min. After incubation, absorbance was measured at 660 nm (A2). Trolox equivalent (1 mmol/L) was used as the standard solution. Measurements were performed in triplicate and the changes in absorbance (∆Abs) were calculated. TAS levels were expressed in mmol Trolox Eq/L [28].

Oxidative stress index (OSI): OSI was calculated as the ratio of TOS to TAS and expressed as a percentage. TAS levels were converted to µmol units for this calculation and the results were reported in Arbitrary Units (AU) [28].

Paraoxonase-1 (PON-1) activity: The spectrophotometric measurement of paraoxonase activity was conducted using commercially available kits (RelAssay Diagnostic^®^, Turkey). The rate of paraoxonase hydrolysis (diethyl p-nitrophenylphosphate in 50 mM glycine/NaOH, pH 10.5 containing 1 mM CaC_l2_) was determined by measuring the increase in absorption at 412 nm at 37 °C. The amount of generated p-nitrophenol was determined from the molar absorption coefficient at pH 8.5, which was 18.290 M^−1^cm^−1^ at pH 10.5. The quantity of enzyme required to catalyze the hydrolysis of one μmol of substrate at 37 °C was defined as one enzyme unit (U/L) [29]. 

### 2.7. Histological Analysis

Heart and lung tissue samples were fixed in 10% neutral-buffered formalin. Following fixation, tissue specimens were dehydrated through an increasing grade series of ethanol, cleared in xylene, and embedded in paraffin. Sections of 4 µm thickness were cut from paraffin tissue blocks using a microtome (Leica RM2245, Nussloch, Germany). For histopathological evaluations, sections were stained with hematoxylin and eosin (H&E). All stained sections were examined under a light microscope (Leica DM 4000 B, Germany) in a blinded manner, and micrographs of specimens were captured using the Leica LAS V4.12 software.

Histopathological changes in the lung tissue were evaluated based on a previous study by Kao et al. [30]. Briefly, lung specimens were examined under 200× and 400× magnifications, and lung injury was assessed in 10 randomly selected non-overlapping fields considering the alveolar wall edema, hemorrhage, vascular congestion, and polymorphonuclear leukocyte infiltration. All these parameters were scored from 0 (normal) to 5 (severe) points, and a total injury score ranging between 0 and 20 (0–5, normal or minimal injury; 6–10, mild injury; 11–15, moderate injury; and 16–20, severe injury) was determined by the sum of each score.

H&E-stained myocardial specimens were examined under 200× and 400× magnifications, and myocardial injury was evaluated semi-quantitatively [31,32]. Parameters including microscopic hemorrhage, edema, polymorphonuclear leukocyte infiltration, eosinophil infiltration, loss of striation, and necrosis were scored in a manner described by Papoutsidakis et. al. [31]. Each parameter was assessed in approximately ten fields, and the total injury score for each specimen was calculated by summing the scores of each parameter.

### 2.8. Statistical Analysis

The study data were analyzed using SPSS (Statistical Package for the Social Sciences) 22 (IBM Corp, Armonk, NY, USA). The normality of the variables was analyzed using visual (histogram and probability graphs) and analytical methods (Kolmogorov–Smirnov/Shapiro–Wilk tests). The results are presented as the mean ± standard error. Data were analyzed using the one-way ANOVA. Significant variables were analyzed using the Bonferroni test. For statistical significance, a total type I error level of 5% was used, and statistical significance was set at *p* < 0.05.

## 3. Results

### 3.1. Results of Histopathological Evaluation of Heart Tissue Samples

The necrosis, polymorphonuclear leukocytes, eosinophils, edema, and total damage scores were significantly different between the groups (*p* = 0.001, *p* < 0.0001, *p* = 0.028, *p* < 0.0001, *p* < 0.0001, respectively) (Table 1).

Necrosis was more prominent in the DIR, DIR-S, and DIR-FS groups than in group C (*p* < 0.0001, *p* = 0.010, and *p* = 0.014, respectively) as well as being greater in the DIR group in comparison to group D. Necrosis was less in the DIR-S, DIR-F, and DIR-FS groups than in the DIR group (*p* = 0.020, *p* = 0.002, and *p* = 0.009, respectively) (Table 1). 

Polymorphonuclear leukocyte infiltration was more extensive in the DIR, DIR-S, DIR-F, and DIR-FS groups compared to groups C (*p* < 0.0001, all) and D (*p* < 0.0001, *p* < 0.0001, *p* = 0.010, and *p* = 0.001, respectively). Group DIR-F had less polymorphonuclear leukocyte infiltration compared to the DIR group (*p* = 0.024) (Table 1).

Level of eosinophil infiltration was found to be significantly higher in group DIR compared to groups C and D (*p* = 0.005 and *p* = 0.005, respectively). The presence of eosinophil was significantly reduced in the DIR-S, DIR-F, and DIR-FS groups compared with the DIR group (*p* = 0.006, *p* = 0.005, and *p* = 0.005, respectively) (Table 1).

Edema was more extensive in all groups compared to group C (*p* < 0.0001, all). Additionally, edema was significantly greater in the DIR and DIR-S groups compared to group D (*p* < 0.0001 and *p* = 0.002, respectively). However, edema was significantly reduced in the DIR-F and DIR-FS groups compared to the DIR group (*p* = 0.004 and *p* = 0.015, respectively) (Table 1).

No significant differences were found between the groups in terms of loss of striation and microscopic hemorrhage (Table 1).

The total injury score was significantly higher in all groups compared to group C (*p* < 0.0001, all). Additionally, the total injury scores of the DIR, DIR-S, and DIR-FS groups were significantly higher than that of group D (*p* < 0.0001, *p* = 0.004, and *p* = 0.022, respectively). Additionally, the total injury score was significantly lower in the DIR-S, DIR-F, and DIR-FS groups than in the DIR group (*p* = 0.005, *p* < 0.0001, and *p* < 0.0001, respectively) (Table 1) (Figure 1, Figure 2, Figure 3, Figure 4, Figure 5 and Figure 6).

### 3.2. Results of Histopathological Evaluation of Lung Tissue Samples

Alveolar wall edema, hemorrhage, vascular congestion, polymorphonuclear leukocyte infiltration, and total lung injury score were found to be significantly different between the groups (*p* < 0.0001, *p* = 0.001, *p* < 0.0001, *p* < 0.0001, and *p* < 0.0001 respectively) (Table 2).

Alveolar wall edema was present across all experimental groups as opposed to the control group (*p* = 0.004, *p* < 0.0001, *p* < 0.0001, *p* = 0.003, and *p* < 0.0001, respectively). Notably, alveolar wall edema exhibited heightened prominence in the DIR and DIR-S groups compared to that in group D (*p* < 0.0001 and *p* = 0.034, respectively), whereas a mitigated level of alveolar wall edema was discerned in the DIR-F and DIR-FS groups relative to the DIR group (*p* < 0.0001 and *p* = 0.006, respectively) (Table 2).

The DIR group exhibited considerable alveolar hemorrhage compared to the C and D groups (*p* < 0.0001, respectively). However, there was a reduced occurrence of alveolar hemorrhage in the DIR-S, DIR-F, and DIR-FS groups compared with the DIR group (*p* = 0.007, *p* < 0.0001 and *p* = 0.002, respectively) (Table 2).

Vascular congestion was more extensive in the DIR, DIR-S, and DIR-FS groups than in the control group (*p* < 0.0001, *p* = 0.001 and *p* = 0.024, respectively). Also, vascular congestion in the DIR and DIR-S groups was considerably greater compared to the D group (*p* < 0.0001 and *p* = 0.026, respectively). Vascular congestion was notably less in the DIR-S, DIR-F, and DIR-FS groups compared to that in the DIR group (*p* = 0.015, *p* < 0.0001, and *p* < 0.0001, respectively) (Table 2).

Polymorphonuclear leukocyte infiltration was significantly elevated in all experimental groups compared to that in the control group (*p* = 0.007, *p* < 0.0001, *p* < 0.0001, *p* < 0.0001, and *p* < 0.0001, respectively). Group DIR exhibited heightened polymorphonuclear leukocyte infiltration compared to group D (*p* < 0.0001), whereas a discernibly reduced level of infiltration was evident in the DIR-S, DIR-F, and DIR-FS groups relative to the DIR group (*p* = 0.006, *p* < 0.0001, and *p* = 0.003, respectively) (Table 2)

The lung injury scores of all the experimental groups were significantly higher than that of the control group (*p* < 0.0001, all). Moreover, lung injury in the DIR, DIR-S, and DIR-FS groups was more severe relative to the D group (*p* < 0.0001, *p* < 0.0001, and *p* = 0.023, respectively), whereas lung injury in the DIR-S, DIR-F, and DIR-FS groups was notably attenuated compared to the DIR group (*p* < 0.0001, all) (Table 2) (Figure 7, Figure 8, Figure 9, Figure 10, Figure 11 and Figure 12).

### 3.3. Heart Tissue Biochemistry Results

When the groups were compared in terms of TOS, TAS, OSI, and PON-1 parameters in heart tissue, significant differences were found in all parameters (*p* = 0.011, *p* = 0.045, *p* < 0.0001, and *p* < 0.0001, respectively). (Table 3)

The TOS levels were significantly higher in the DIR group than in the C and D groups (*p* < 0.0001 and *p* = 0.004, respectively). In addition, TOS levels were significantly lower in the DIR-F and DIR-FS groups than in the DIR group (*p* = 0.004 and *p* = 0.005, respectively) (Table 3).

The TAS levels were significantly lower in the DIR group than in the C and D groups (*p* = 0.007 and *p* = 0.031, respectively). The TAS levels were also significantly lower in the DIR group than in the D group (*p* = 0.012) (Table 3). 

The OSI levels were significantly higher in the DIR and DIR-S groups than in the C group (*p* < 0.0001 and *p* < 0.0001, respectively). In addition, OSI was significantly lower in the DIR-S, DIR-F, and DIR-FS groups than in the DIR group (*p* = 0.006, *p* < 0.0001, *p* < 0.0001, and *p* < 0.0001, respectively) (Table 3).

PON-1 enzyme activity was significantly higher in the DIR, DIR-S, DIR-F, and DIR-FS groups than in the C group (*p* < 0.0001, *p* = 0.029, *p* < 0.0001, *p* < 0.0001, and *p* < 0.0001, respectively). Similarly, PON-1 enzyme activity was significantly increased in the DIR, DIR-S, and DIR-FS groups compared to that in the control group (*p* < 0.0001, *p* < 0.0001, *p* < 0.0001, and *p* < 0.0001, respectively). In addition, PON-1 enzyme activity levels were significantly lower in the DIR-S, DIR-F, and DIR-FS groups than in the DIR group (*p* = 0.037, *p* < 0.0001, and *p* = 0.007, respectively) (Table 3).

### 3.4. Lung Tissue Biochemistry Results

When the groups were compared in terms of TOS, TAS, OSI, and PON-1 parameters in lung tissue, significant differences were found in all parameters (*p* < 0.0001).

TAS levels were significantly lower in the DIR, DIR-S, and DIR-FS groups than in the C group (*p* < 0.0001, *p* = 0.002, and *p* = 0.008, respectively). Similarly, TAS levels were significantly lower in the DIR and DIR-S groups than in group D (*p* < 0.0001 and *p* = 0.027, respectively). In addition, the TAS levels were significantly higher in the DIR group than in the DIR-F and DIR-FS groups (*p* = 0.004 and *p* = 0.035, respectively) (Table 4).

TOS levels were significantly higher in the DIR and DIR-S groups than in group C (*p* < 0.0001 and *p* = 0.027, respectively). Similarly, the TOS levels were significantly higher in the DIR group than in the D group (*p* < 0.000). In addition, a significant decrease in TOS levels was found in the DIR-S, DIR-F, and DIR-FS groups compared to the DIR group (*p* < 0.001, *p* = 0.002, and *p* < 0.0001, respectively) (Table 4).

OSI levels were significantly increased in the DIR, DIR-S, and DIR-FS groups compared to those in group C (*p* < 0.0001, *p* = 0.017, and *p* = 0.039, respectively). Similarly, the OSI level was significantly higher in group DIR than in group D (*p* < 0.0001). In addition, a significant decrease in OSI was observed in the DIR-S, DIR-F, and DIR-FS groups compared to group DIR (all *p* < 0.0001) (Table 4).

PON-1 enzyme activity was significantly higher in the DIR group than in the C and D groups (all *p* < 0.0001). In addition, PON-1 enzyme activity levels were significantly lower in the DIR-S, DIR-F, and DIR-FS groups than in group DIR (*p* = 0.009, *p* < 0.0001, *p* < 0.0001, *p* < 0.0001, respectively) (Table 4).

## 4. Discussion

In this study, we established an animal model of lower-extremity ischemia–reperfusion injury and demonstrated that fullerenol C60 and sevoflurane have protective effects on the heart and lungs of diabetic mice. Fullerenol C60 and sevoflurane ameliorated histopathological changes and alleviated oxidative parameters, leading to an overall improvement in the lungs and heart. 

Ischemia–reperfusion injury poses a significant risk to various organs, including the heart, lungs, kidneys, and brain, by exacerbating oxidative stress through the generation of reactive oxygen species (ROS) and subsequent lipid peroxidation [3,19,20,21]. Patients with diabetes mellitus are thought to be susceptible to IRI due to underlying microvascular endothelial dysfunction, characterized by conditions like capillary basement membrane thickening and endothelial hyperplasia, which impair tissue oxygenation and exacerbate hypoxic conditions [33,34]. Furthermore, diabetes reduces the clearance of ROS, exacerbating oxidative stress. Hyperglycemia also decreases antioxidant capacity by reducing the activities of superoxide dismutase enzyme 2, glutathione peroxidase, and catalase [24]. 

Our study builds upon existing research demonstrating the protective effects of fullerenol C60 and sevoflurane in ischemia–reperfusion scenarios. In studies investigating cerebral ischemia–reperfusion injury, fullerene C60 has been shown to reduce the development of cerebral edema and infarct area, decrease the neural defect development score, and significantly suppress interleukin 6 and matrix metalloproteinase 9 RNA expression [35,36]. Paralleling the findings on cerebral edema in the literature, we observed a significant increase in interstitial edema caused by ischemia–reperfusion injury in cardiac tissue across all groups compared with the control group.

A previous study has demonstrated that fullerene C60 offers a protective effect against lung ischemia–reperfusion injury. It reduced pulmonary artery and vein pressures and lung weight compared with the ischemia group [37]. Our study further demonstrated that fullerene C60 effectively decreased pulmonary edema, hemorrhage, vascular congestion, and polymorphonuclear leukocyte infiltration in lung tissue during limb ischemia–reperfusion injury.

Fullerenol C60 is a powerful protector against ischemia–reperfusion injury that effectively scavenges free radicals. When applied to small intestinal ischemia–reperfusion, alkenes and malondialdehyde (MDA) levels were significantly decreased, while glutathione levels were increased in the group treated with fullerenol [38]. Furthermore, farboxyfullerene decreased the increase in lipid peroxidation products, decreased MDA levels, and increased superoxide dismutase, glutathione, and catalase levels depleted by ischemia in cerebral ischemia–reperfusion injury [39,40,41,42]. Our study demonstrated the protective effect of fullerene by showing that oxidant status parameters decreased in both heart and lung tissues in the fullerene-treated groups compared to the ischemia–reperfusion group. This was evidenced by the reduction in TOS, OSI, and PON-1 levels.

Sevoflurane, commonly used in clinical anesthesia, has been reported to have beneficial effects on the heart and lungs in ischemia–reperfusion injury [43,44]. Studies have shown that sevoflurane can improve lung injury caused by endotoxins and alleviate endotoxin-induced lung edema and inflammatory cell infiltration without adversely affecting normal lung tissue [45]. It activates Nrf2/ARE signaling and inhibits inflammatory infiltration and the release of proinflammatory cytokines via the improvement of antioxidant defenses and the maintenance of redox homeostasis [46]. 

The effect of sevoflurane on myocardial ischemia has been shown in both in vivo and in vitro models, and is associated with anti-inflammation and the inhibition of inflammasomes, IL-1β, IL-18, and cell pyroptosis [47,48]. However, among the potential molecular mechanisms responsible for the sevoflurane-induced improvement of cardiac function, pathway signaling seems to be focused on the structure and function of mitochondria. Yu et al. demonstrated that sevoflurane protects rat hearts against ischemia–reperfusion injury by ameliorating mitochondrial impairment, oxidative stress, and rescuing autophagic clearance. Their study observed the prevention of mitochondrial destruction and an increase in ATP content in sevoflurane-treated hearts. Sevoflurane administration significantly increased the transcriptional levels of genes related to mitochondrial function (Cycs, Cox4il, Ndufa2, Ndufa4, Ndufa8, Cox7a1, Cox7a2, and TFAM), and the expressions of mitochondrial proteins (Nrf-1 and PGC-1α) were significantly upregulated after sevoflurane administration. Their results indicate that the cardiac protection offered by sevoflurane involves restoring mitochondrial bioenergetic metabolism and autophagosome clearance, inhibiting excessive ROS production, ameliorating protein aggregation, activating class I PI3K, downregulating the expression of class III PI3K, and suppressing autophagy (mitophagy) at the end of reperfusion [10].

However, it is important to acknowledge several limitations of our study. Firstly, the sample size was constrained by ethical guidelines, affecting statistical power. Secondly, diabetes status was defined solely by serum glucose levels due to resource limitations, precluding a histological examination of pancreatic tissue.

## 5. Conclusions

In conclusion, the findings from our histological examinations underscore the significant protective effects of sevoflurane and fullerenol C60 against ischemia–reperfusion injury in heart and lung tissues. The observed reduction in PMNL infiltration, necrotic cardiomyocytes, alveolar hemorrhage, interstitial and alveolar edema, and vascular congestion, along with lower total damage scores, highlights the efficacy of these treatments. Additionally, the decreased levels of TAS, TOS, OSI, and PON-1 further corroborate the antioxidative and protective properties of sevoflurane and fullerenol C60. 

These findings suggest that sevoflurane and fullerenol C60 have significant potential for clinical application in the management of ischemia–reperfusion injury. This potential is particularly relevant in tourniquet-assisted limb surgery, peripheral arterial bypass surgery, or as a primary treatment for peripheral arterial disease in diabetic patients. The promising results from this study warrant further research and development. Further exploration and validation of the broader clinical benefits of these agents are necessary to confirm their effectiveness.

## Figures and Tables

**Figure 1 medicina-60-01232-f001:**
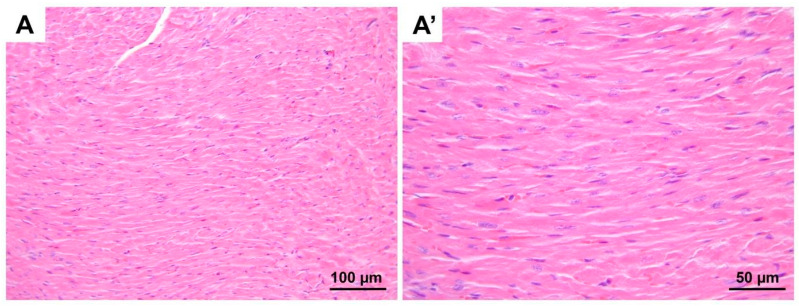
Representative micrographs of H&E-stained heart sections from control group. Cardiomyocytes with normal euchromatic nuclei and similar cytoplasmic staining properties are seen. (**A**) 200× magnification; (**A′**) 400× magnification. H&E, hematoxylin and eosin stain.

**Figure 2 medicina-60-01232-f002:**
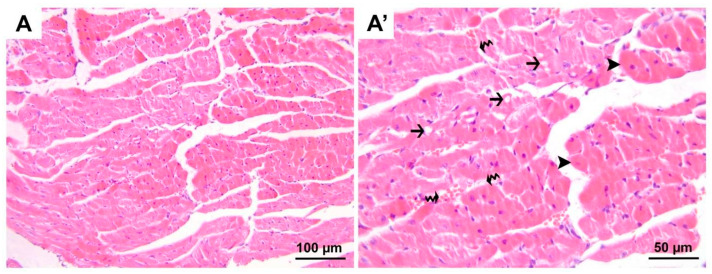
Representative micrographs of H&E-stained heart sections from group D. Heterogenous appearances throughout the cardiomyocytes were noted to vary in dimension, being either atrophic or hypertrophic (black arrow heads) and in staining properties, being hyper-eosinophilic in some regions. Additionally, cardiomyocytes exhibiting varying degrees of vacuolization (black arrows) were seen. Also, interstitial edema (black wavy arrows) expanding the distance between neighboring cardiomyocytes was accompanied by erythrocyte extravasation in a few regions. (**A**) 200× magnification; (**A′**), 400× magnification. H&E, hematoxylin and eosin stain.

**Figure 3 medicina-60-01232-f003:**
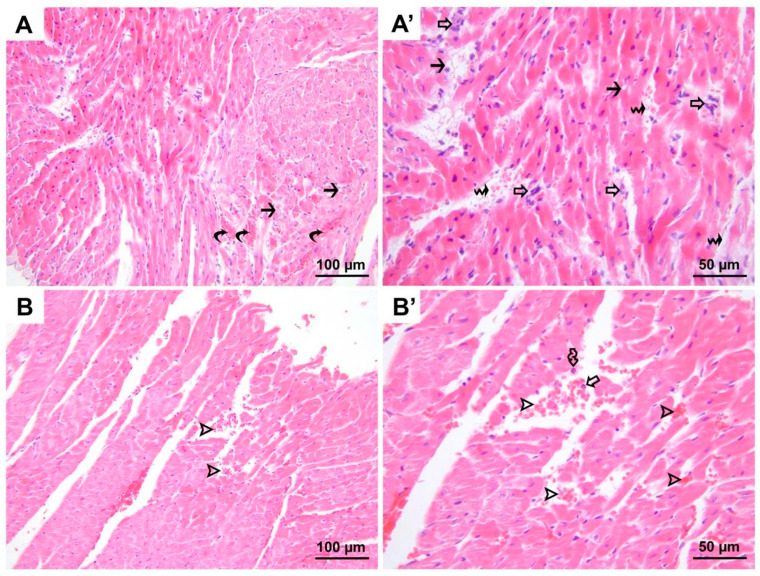
Representative micrographs of H&E-stained heart sections from group DIR. Besides the hyper-eosinophilia, vacuolization of varying degrees (black arrows, (**A**,**A′**)) is present in cardiomyocytes. Microscopic bleeding foci (hollow arrow heads, (**B**,**B′**)) are noted in addition to congestion (black curved arrows, (**A**)). Also, prominent polymorphonuclear leukocyte infiltration (hollow arrows, (**A’**,**B′**)) is seen in some regions in contrast to groups C and D. Interstitial edema (black wavy arrows, (**A′**)), occasionally accompanied by erythrocyte extravasation, is more widespread and severe, and eosinophils (hollow wavy arrow, (**B′**)) are also present in the lesion area along with neutrophils. (**A**,**B**), 200× magnification; (**A′**,**B′**), 400× magnification. H&E, hematoxylin and eosin stain.

**Figure 4 medicina-60-01232-f004:**
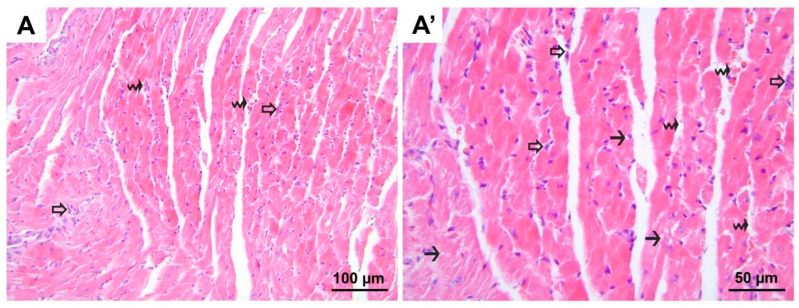
Representative micrographs of H&E-stained heart sections from group DIR-S. Besides the hyper-eosinophilia, degenerative changes characterized by vacuolization (black arrows, (**A′**)) are observed in cardiomyocytes. Interstitial edema (black wavy arrows, (**A**,**A′**)), which may occasionally be accompanied by erythrocyte extravasation or polymorphonuclear leukocyte infiltration, is noted, and isolated polymorphonuclear leukocyte infiltration foci (hollow arrows, (**A**,**A′**)) are observed. (**A**) 200× magnification; (**A′**) 400× magnification. H&E, hematoxylin and eosin stain.

**Figure 5 medicina-60-01232-f005:**
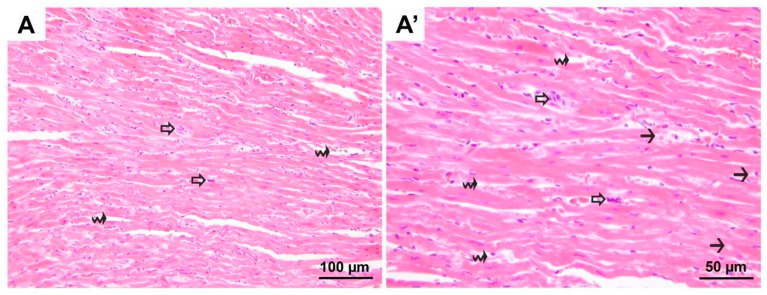
Representative micrographs of H&E-stained heart sections from group DIR-F. Cardiomyocytes with vacuolar degeneration (black arrows, (**A′**)), polymorphonuclear leukocyte infiltration (hollow arrows, (**A**,**A′**)) and interstitial edema (black wavy arrows, (**A**,**A′**)) were relatively less compared to other groups that underwent ischemia–reperfusion. (**A**) 200× magnification; (**A′**) 400× magnification. H&E, hematoxylin and eosin stain.

**Figure 6 medicina-60-01232-f006:**
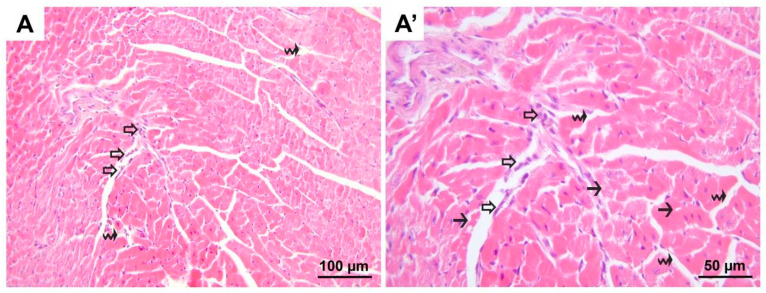
Representative micrographs of H&E-stained heart sections from group DIR-FS. It is noteworthy that vacuolization in cardiomyocytes (black arrows, (**A′**)), polymorphonuclear leukocyte infiltration (hollow arrows, (**A**,**A′**)), and interstitial edema (black wavy arrows, (**A**,**A′**)) were all milder compared to the DIR and DIR-S groups. (**A**) 200× magnification; (**A′**) 400× magnification. H&E, hematoxylin and eosin stain.

**Figure 7 medicina-60-01232-f007:**
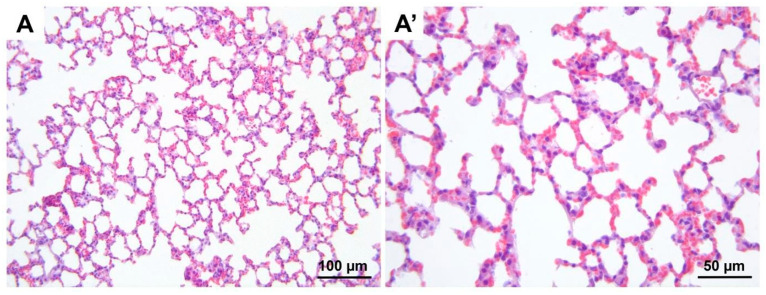
Representative micrographs of H&E-stained lung sections from control group. (**A**) 200× magnification; (**A′**) 400× magnification. H&E, hematoxylin and eosin stain.

**Figure 8 medicina-60-01232-f008:**
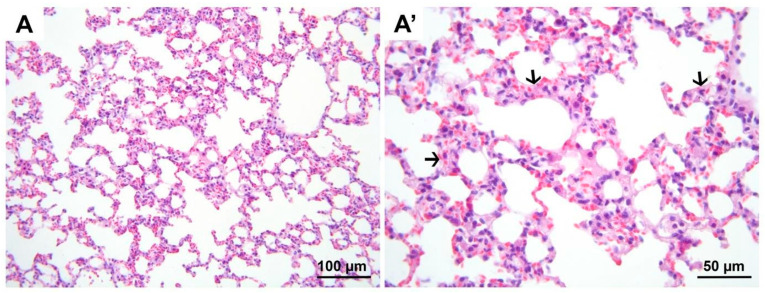
Representative micrographs of H&E-stained lung sections from group D. Relatively thicker alveolar wall (black arrows, (**A′**)) compared to the control group is seen. (**A**) 200× magnification; (**A′**) 400× magnification. H&E, hematoxylin and eosin stain.

**Figure 9 medicina-60-01232-f009:**
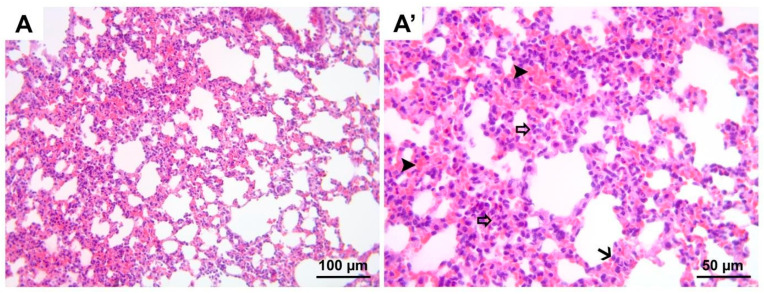
Representative micrographs of H&E-stained lung sections from group DIR. Compared to all groups, significant hemorrhage (black arrow heads, (**A′**)), polymorphonuclear leukocyte infiltration (hollow arrows, (**A′**)), and alveolar wall edema (black arrow, (**A′**)) are noted. (**A**) 200× magnification; (**A′**) 400× magnification. H&E, hematoxylin and eosin stain.

**Figure 10 medicina-60-01232-f010:**
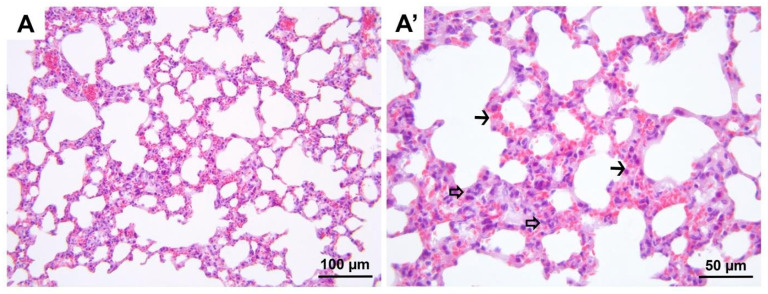
Representative micrographs of H&E-stained lung sections from group DIR-S. Although significant alveolar wall edema (black arrows, (**A′**)) is observed compared to the control and D groups, less polymorphonuclear leukocyte infiltration (hollow arrows, (**A′**)) is observed compared to the DIR group. (**A**) 200× magnification; (**A′**) 400× magnification. H&E, hematoxylin and eosin stain.

**Figure 11 medicina-60-01232-f011:**
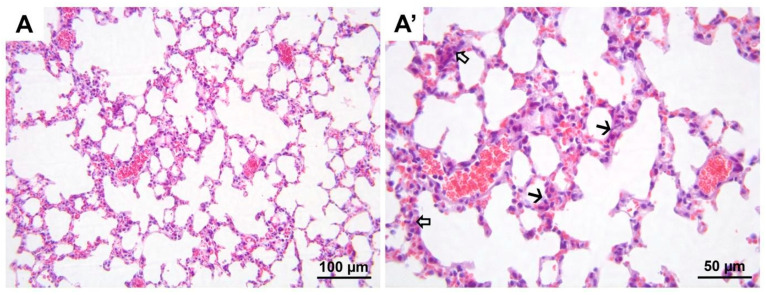
Representative micrographs of H&E-stained lung sections from group DIR-F. Significantly less alveolar wall edema (black arrows, (**A′**)) and polymorphonuclear leukocyte infiltration (hollow arrows, (**A′**)) are seen compared to the DIR group. (**A**) 200× magnification; (**A′**) 400× magnification. H&E, hematoxylin and eosin stain.

**Figure 12 medicina-60-01232-f012:**
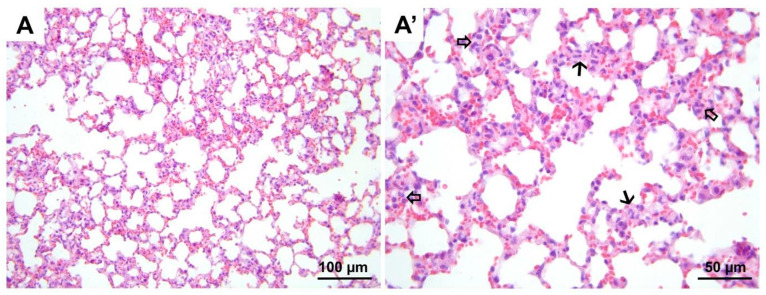
Representative micrographs of H&E-stained lung sections from group DIR-FS. Alveolar wall edema (black arrows, (**A′**)) and polymorphonuclear leukocyte infiltration (hollow arrows, (**A′**)) appear to be significantly less compared to the DIR group. (**A**) 200× magnification; (**A′**) 400× magnification. H&E, hematoxylin and eosin stain.

**Table 1 medicina-60-01232-t001:** Results of histopathological evaluation of heart tissue samples [Mean ± SE].

	Group C (*n* = 6)	Group D (*n* = 7)	Group DIR (*n* = 7)	Group DIR-S (*n* = 7)	Group DIR-F (*n* = 7)	Group DIR-FS (*n* = 7)	*p* **
Necrosis	0.57 ± 0.30	1.43 ± 0.37	3.00 ± 0.00 *^,&^	1.83 ± 0.48 *^,+^	1.43 ± 0.20 ^+^	1.71 ± 0.36 *^,+^	0.001
Polymorphonuclear leukocytes	0.00 ± 0.00	0.43 ± 0.20	2.33 ± 0.33 *^,&^	2.00 ± 0.26 *^,&^	1.43 ± 0.27 *^,&,+^	1.72 ± 0.28 *^,&^	<0.0001
Eosinophils	0.00 ± 0.00	0.00 ± 0.00	0.33 ± 0.21 *^,&^	0.00 ± 0.00 *^+^*	0.00 ± 0.00 *^+^*	0.00 ± 0.00 *+*	0.028
Loss of striation	0.00 ± 0.00	0.00 ± 0.00	0.00 ± 0.00	0.00 ± 0.00	0.00 ± 0.00	0.00 ± 0.00	-
Edema	0.14 ± 0.14	1.00 ± 0.22 *	2.33 ± 0.21 *^,&^	2.00 ± 0.26 *^,&^	1.43 ± 0.20 *^,+^	1.57 ± 0.20 *^,+^	<0.0001
Microscopic hemorrhage	0.14 ± 0.14	0.43 ± 0.20	0.83 ± 0.31	0.50 ± 0.22	0.71 ± 0.18	0.57 ± 0.20	0.285
Total injury score	0.86 ± 0.34	3.86 ± 0.63 *	8.83 ± 0.70 *^,&^	6.33 ± 0.56 *^,&,+^	5.00 ± 0.58 *^,+^	5.71 ± 0.52 *^,&,+^	<0.0001

*p* **: Significance level by ANOVA test, *p* < 0.05; * *p* < 0.05: compared with group C; ^&^ *p* < 0.05: compared with group D; ^+^ *p* < 0.05: compared with group DIR.

**Table 2 medicina-60-01232-t002:** Results of histopathological evaluation of lung tissue samples [Mean ± SE].

	Group C (*n* = 6)	Group D (*n* = 7)	Group DIR (*n* = 7)	Group DIR-S (*n* = 7)	Group DIR-F (*n* = 7)	Group DIR-FS (*n* = 7)	*p* **
Alveolar wall edema	0.67 ± 0.23	1.63 ± 0.24 *	2.82 ± 0.22 *^,&^	2.33 ± 0.20 *^,&^	1.66 ± 0.27 *^,+^	1.87 ± 0.13 *^,+^	<0.0001
Hemorrhage	0.26 ± 0.10	0.42 ± 0.18	1.48 ± 0.30 *^,&^	0.70 ± 0.20 *^,+^	0.47 ± 0.14 ^+^	0.61 ± 0.16 ^+^	0.001
Vascular congestion	0.77 ± 0.12	1.04 ± 0.17	2.20 ± 0.11 *^,&^	1.58 ± 0.17 *^,&,+^	1.22 ± 0.16 ^+^	1.30 ± 0.21 *^,+^	<0.0001
Polymorphonuclear leukocyte infiltration	1.02 ± 0.13	1.93 ± 0.20 *	3.55 ± 0.41 *^,&^	2.55 ± 0.24 *^,+^	2.20 ± 0.17 *^,+^	2.51 ± 0.19 *^,+^	<0.0001
Total injury score	2.73 ± 0.28	5.02 ± 0.50 *	10.05 ± 0.44 *^,&^	7.17 ± 0.40 *^,&,+^	5.56 ± 0.36 *^,+^	6.30 ± 0.32 *^,&,+^	<0.0001

*p* **: significance level by ANOVA test, *p* < 0.05; * *p* < 0.05: compared with group C; ^&^ *p* < 0.05: compared with group D; ^+^ *p* < 0.05: compared with group DIR.

**Table 3 medicina-60-01232-t003:** Heart tissue oxidant status parameters [Mean ± SE].

	Group C (*n* = 6)	Group D (*n* = 7)	Group DIR (*n* = 7)	Group DIR-S (*n* = 7)	Group DIR-F (*n* = 7)	Group DIR-F-S (*n* = 7)	*p* **
TOS (μmol H_2_O_2_Equiv./L)	1.25 ± 0.37	2.62 ± 0.36	10.06 ± 2.99 *^,&^	5.40 ± 2.50	2.63 ± 0.59 +	2.83 ± 1.05 +	0.011
TAS (mmol Trolox Equiv./L)	0.51 ± 0.10	0.49 ± 0.11	0.23 ± 0.04 *^,&^	0.29 ± 0.04 *	0.40 ± 0.04	0.35 ± 0.02	0.045
OSI	2.26 ± 0.37	6.40 ± 1.47	51.19 ± 15.37 *^,&^	20.48 ± 8.87+	7.26 ± 1.53 ^+^	6.44 ± 1.93 ^+^	<0.0001
PON-1 (U/L)	12.70 ± 0.89	13.77 ± 1.10	28.91 ± 2.85 *^,&^	24.05 ± 0.75 *^,&,+^	17.63 ± 1.95 *^,+^	22.46 ± 0.78 *^,&,+^	<0.0001

*p* **: significance level by ANOVA test, *p* < 0.05; * *p* < 0.05: compared with group C; ^&^ *p* < 0.05: compared with group D; ^+^ *p* < 0.05: compared with group DIR.

**Table 4 medicina-60-01232-t004:** Lung tissue oxidant status parameters [Mean ± SE].

	Group C (*n* = 6)	Group D (*n* = 7)	Group DIR (*n* = 7)	Group DIR-S (*n* = 7)	Group DIR-F (*n* = 7)	Group DIR-F-S (*n* = 7)	*p* **
TOS (μmol H_2_O_2_Equiv./L)	23.97 ± 4.01	27.93 ± 6.19	60.94 ± 6.75 *^,&^	39.12 ± 3.97 *^,+^	30.90 ± 2.35 ^+^	37.22 ± 2.45 ^+^	<0.0001
TAS (mmol Trolox Equiv./L)	1.59 ± 0.04	1.51 ± 0.03	1.24 ± 0.08 *^,&^	1.35 ± 0.04 *^,&^	1.45 ± 0.02 ^+^	1.39 ± 0.03 *^,+^	<0.0001
OSI	15.02 ± 2.43	18.55 ± 4.12	55.09 ± 7.40 *^,&^	28.73 ± 2.14 *^,+^	21.29 ± 1.51 ^+^	26.78 ± 1.82 *^,+^	<0.0001
PON-1 (U/L)	34.80 ± 2.11	34.07 ± 1.07	16.80 ± 3.60 *^,&^	27.28 ± 2.14	35.21 ± 3.51 ^+^	32.08 ± 2.46 ^+^	<0.0001

*p* **: significance level by ANOVA test, *p* < 0.05; * *p* < 0.05: compared with group C; ^&^ *p* < 0.05: compared with group D; ^+^ *p* < 0.05: compared with group DIR.

## Data Availability

The datasets used and/or analyzed during the current study are available from the corresponding author upon reasonable request.

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
