# Peer review of "Effects of Sevoflurane and Fullerenol C60 on the Heart and Lung in Lower-Extremity Ischemia–Reperfusion Injury in Streptozotocin-Induced Diabetes Mice"

_medicina, 2024, doi:10.3390/medicina60081232_

Round 1

Reviewer 1 Report

Comments and Suggestions for Authors

Comments on the Quality of English Language

Author Response

Dear Editor,

We thank the reviewers for their valuable opinions and contributions to our study. We carefully evaluated the valuable reviewer opinions and made the necessary changes to our manuscript according to these suggestions. We indicate the changes made below.

Reviewer #1 (Red)

The article entitled “Effects of Sevoflurane and Fullerenol C60 on the Heart and Lung in Lower Extremity Ischaemia Reperfusion In-jury in Streptozotocin-Induced Diabetes Mice” represents a good experimental work.

Here are my comments

Abstract

  • The background is too short and needs to be rephrased and include some information about the used drugs

Response: Information about Sevoflurane and Fullerenol has been added. The background was not extended further to avoid exceeding the word limit for the abstract.

  • The experimental animal procedures need to be shortened and the detailed animal surgery should be mentioned in the methods section. The mice number here is 46 and in Methods 41 please correct.

Response: The error was made inadvertently and has been corrected in the abstract.

  • The dose of Streptozotocin should be mentioned.

Response:The dose of Streptozotocin has been added to abstract

  • The full description should precede abbreviations (DIR TOS, OSI, and PON) please do it throughout the whole manuscript.

Response:This has been corrected throughout the manuscript.

Introduction

  • The first paragraph needs to be rephrased reviewing more articles about Diabetes Mellitus, pathogenesis, types, complications, epidemiology, and global prevalence of the disease.

Response:First paragragh was rewritten.

  • It was preferable to add transition sentences to prepare the reader like one of the major vascular diabetic complications is ischemia then move to the next section defining it.

Response:Transion sentenses has been added.

  • Again, the full description should precede the abbreviations (ARDS).

Response:Full description of ARDS has been added

  • “The mechanism of the exaggerated response to ischemia-reperfusion injury in diabetes has not been fully elucidated, but it is thought that excessive formation of ROS due to hyperglycemia and depletion of endogenous antioxidant defense mechanisms contribute to the damage” A reference is needed and the author should add more about the hyperglycemia induced oxidative stress, advanced glycated end products and their receptors and signaling pathway that augments oxidative stress, DNA damage, and vascular wall pathology.

Response:Rewritten according to reviewes comment

  • “The objective of this research was to illustrate how fullerenol C₆₀ provides protective effects on the lungs and heart when lower extremity ischemia-reperfusion injury is induced in mice.” Can you make it clearer, like which effects, is it antioxidant effect and on which cells of the lungs and hearts?

Response:Rewritten according to reviewes comment

Materials and Methods

Animals:

  • Sample size calculation was not mentioned please add it.

Response:The maximum number of animals per group was limited by the ethics committee; therefore, a sample size calculation was not performed prior to the study

  • The housing conditions of the mice should be mentioned (Cages, diet, humidity,…etc.)

Response:Housing conditions have been added

  • The group's abbreviations (DIR, DIR F,…..etc.) better be added from the first at the experimental setup.

Response:Group's abbreviations have been addedd under the Experimental Setup and Animals section

  • The dose of streptozotocin is better to be supplemented by a reference. Maker is not mentioned.

Response:The streptozotocin dose was selected in accordance with our previous studies and literature. A reference has been added, and the maker has been specified.

  • Please mention the makers for all and revise it through the methods section. How the blood glucose was measured (mention the equipment and the maker).

Response:The glucometer's brand has been specified, and it was mentioned that blood glucose was measured from the tail vein.

  • Why the authors use 4 weeks to establish chronic diabetic complications? (Add a reference)

Response:The 4-week period was selected based on our previous studies and literature. A reference has been added to the main manuscript.

  • How did you confirm vascular complications? Did you do a pilot experiment?

Response:As similar models were not used in our previous studies, no pilot experiment was conducted. References have been included in the main manuscript.

  • Tissue collection and analysis
  • How tissues and blood were collected and preserved for biochemical assays? Please specify in the methods section.

Response:Details have been added to the methods section to specify how tissues and blood were collected and preserved for biochemical assays.

  • Please mention details about the histopathological and procedure and add references.

Response:Details have been added about the histopathological and biochemical assays

  • The authors did not mention how the tissues were homogenized and the fasting stats of the animals.

Response:Information on tissue homogenization and the fasting status of the animals has been added.

  • Please add the product number and specify the machines used in the acquisition with their maker, and country.

Response:Product numbers, along with the details of the machines used (including maker and country), have been added to the relevant sections.

  • Oxidative Stress Index (OSI): Add reference.

Response:A reference has been added for the Oxidative Stress Index (OSI).

Statistical analysis

  • Please add the maker to the software used.

Response:Added

Results

  • It needs to be rewritten and organized to present the current work

Response: editing

  • It would be better if the author started by the establishment of a chronic diabetic model with H&E of the pancreas and peripheral arteries as well as fasting blood glucose.

Response: We thank the reviewer for their valuable suugestions and contributions the study. However, due to the practical and financial constrains. It is nat feasible to redo the entire study. We fill consider this recommendation for future research.

Since the results presented in tables (1, and 2) are (scores), the authors would have used a non-parametric test like the Kruskal-Wallis H test instead of ANOVA

Response: Thank you for your valuable contributions.

  • In Tables 3, and 4 the units are not mentioned, the index (OSI) is a non-parametric data it would be preferable to use a non-parametric ANOVA. PON should be changed into PON-1 in all tables and manuscript.

Response: Editing

Discussion

  • Once more, all abbreviations should be preceded by full names (IL6, MMP, GSH…….etc.)

Response: All abbreviations have now been preceded by their full names throughout the document.

  • Line 68, please rephrase.

Response: Line 68 has been rewritten for clarity.

  • Lines 73-79

Response: Rewritten to improve clarity and content.

  • Please explain why patients with diabetes are at heightened risk of vascular injuries and ischemia.

Response: An explanation has been added to elucidate why diabetic patients are at an increased risk of vascular injuries and ischemia.

  • The discussion is brief and shallow and does not address the molecular mechanisms of Sevoflurane and Fullerenol C60 in the current study.

Response: New paragragh have been added to discussion and it is rewritten

  • The authors also should discuss the adverse effects of both drugs as were as prior research with adverse results

Response: We appreciate your valuable suggestion. However, our study did not investigate the adverse effects of these drugs; such inquiries were beyond the scope of our research objectives. We acknowledge the importance of this aspect and suggest it could be a compelling direction for subsequent research.

  • The authors should discuss their effects on endothelial dysfunction, NO levels, and vascular homeostasis.

Response: Thank you for this insightful observation. Unfortunately, our study did not cover endothelial dysfunction, NO levels, or vascular homeostasis. We recognize the significance of these factors in understanding the comprehensive impacts of the drugs studied and recommend they be addressed in future research endeavors.

Conclusion

  • Must be rewritten to draw a clear conclusion of the current study and provide future recommendations.

Response: The conclusion has been rewritten to clearly summarize the findings of the current study and suggest directions for future research.

References

  • The reviewed literature is limited and includes old articles, DOI is not included.
  • Recent literature including 2024 should be added particularly in the introduction and discussion sections.
  • A thorough English grammar editing, punctuation, and rephrasing are needed to improve your manuscript.

Response: Doi number has been added

Best regards.

Reviewer 2 Report

Comments and Suggestions for Authors

The manuscript is well-structured and addresses an important research question. However, there are areas that need improvement to enhance clarity and readability. The authors should focus on condensing and clarifying the abstract and introduction, providing justifications for methodological choices, organizing the results more effectively, and discussing limitations and broader implications in the conclusion.

Abstract

Strengths:

The abstract provides a clear and informative overview of the research objective, methodology, results, and conclusion.

It effectively highlights the significance of studying the effects of Sevoflurane and Fullerenol C60 in the context of ischemia-reperfusion injury in diabetic mice.

Weaknesses:

The abstract is somewhat verbose, potentially overwhelming the reader with too much detail.

There needs to be a clearer separation between the methods and results sections within the abstract.

Suggestions for Improvement:

Condense the abstract to include only the most critical information, aiming for brevity while maintaining clarity.

Create a more distinct separation between the methods and results to improve readability.

Introduction

Strengths:

The introduction provides a comprehensive background on diabetes mellitus and ischemia-reperfusion injury, establishing the study's relevance.

It introduces the research question and the rationale for investigating Sevoflurane and Fullerenol C60 effectively.

Weaknesses:

The introduction could be more engaging with a stronger opening sentence to capture the reader's interest.

The literature review is thorough but could be streamlined to avoid redundancy.

Suggestions for Improvement:

Start with an intriguing statement or fact to engage the reader from the beginning.

Consolidate the literature review to focus on the most relevant studies, ensuring a smoother narrative flow.

Materials and Methods

Strengths:

The experimental setup is described in detail, including the induction of diabetes and the grouping of mice.

The procedures for ischemia-reperfusion and the administration of Sevoflurane and Fullerenol C60 are clearly explained.

Weaknesses:

Some sections are overly detailed and could benefit from more concise language.

The methodology lacks a clear justification for the chosen dosages of Fullerenol C60 and Sevoflurane.

Suggestions for Improvement:

Simplify descriptions without omitting essential details to enhance readability.

Provide a rationale for the selected dosages to strengthen the methodological rigor and clarity.

Results

Strengths:

The results section presents histological and biochemical findings comprehensively, supported by appropriate figures and tables.

There is a clear comparison between different experimental groups, aiding in the interpretation of the data.

Weaknesses:

The section is dense and may be challenging for readers to follow.

Some interpretations are embedded in the results, which should be reserved for the discussion section.

Suggestions for Improvement:

Use subheadings to organize the results, making the section more reader-friendly.

Move interpretative comments to the discussion section to maintain a clear distinction between findings and their implications.

Discussion

Strengths:

The discussion thoroughly interprets the findings, linking them to the study's objectives and existing literature.

The potential mechanisms underlying the observed effects of Sevoflurane and Fullerenol C60 are effectively discussed.

Weaknesses:

The discussion is somewhat repetitive, reiterating points already made in the results section.

There is limited discussion on the study's limitations and future research directions.

Suggestions for Improvement:

Avoid repetition by summarizing key points more succinctly.

Include a section on the study's limitations and propose specific areas for future research.

Conclusion

Strengths:

The conclusion succinctly summarizes the main findings and their significance.

It highlights the potential clinical applications of the findings.

Weaknesses:

The conclusion does not sufficiently address the broader implications of the study.

There is no mention of any limitations or caveats.

Suggestions for Improvement:

Expand the conclusion to discuss the broader implications of the research in the context of diabetes treatment and ischemia-reperfusion injury management.

Mention any study limitations and suggest future research directions to provide a balanced view.

Comments on the Quality of English Language

ok

Author Response

Dear Editor,

We thank the reviewers for their valuable opinions and contributions to our study. We carefully evaluated the valuable reviewer opinions and made the necessary changes to our manuscript according to these suggestions. We indicate the changes made below.

Reviewer #1 (Red)

The manuscript is well-structured and addresses an important research question. However, there are areas that need improvement to enhance clarity and readability. The authors should focus on condensing and clarifying the abstract and introduction, providing justifications for methodological choices, organizing the results more effectively, and discussing limitations and broader implications in the conclusion.

Abstract

Strengths: The abstract provides a clear and informative overview of the research objective, methodology, results, and conclusion. It effectively highlights the significance of studying the effects of Sevoflurane and Fullerenol C60 in the context of ischemia-reperfusion injury in diabetic mice.

Weaknesses: The abstract is somewhat verbose, potentially overwhelming the reader with too much detail. There needs to be a clearer separation between the methods and results sections within the abstract.

Suggestions for Improvement:

  • Condense the abstract to include only the most critical information, aiming for brevity while maintaining clarity.
  • Create a more distinct separation between the methods and results to improve readability.

Response: The abstract has been revised and rewritten.

Introduction

Strengths: The introduction provides a comprehensive background on diabetes mellitus and ischemia-reperfusion injury, establishing the study's relevance. It introduces the research question and the rationale for investigating Sevoflurane and Fullerenol C60 effectively.

Weaknesses: The introduction could be more engaging with a stronger opening sentence to capture the reader's interest. The literature review is thorough but could be streamlined to avoid redundancy.

Suggestions for Improvement:

  • Start with an intriguing statement or fact to engage the reader from the beginning.

Response: Introduction has been rewritten

  • Consolidate the literature review to focus on the most relevant studies, ensuring a smoother narrative flow.

Response: New referecenses has been added according to reviewers comment and introduciton has been rewritten.

Materials and Methods

Strengths: The experimental setup is described in detail, including the induction of diabetes and the grouping of mice. The procedures for ischemia-reperfusion and the administration of Sevoflurane and Fullerenol C60 are clearly explained.

Weaknesses: Some sections are overly detailed and could benefit from more concise language. The methodology lacks a clear justification for the chosen dosages of Fullerenol C60 and Sevoflurane.

Suggestions for Improvement:

  • Simplify descriptions without omitting essential details to enhance readability.
  • Provide a rationale for the selected dosages to strengthen the methodological rigor and clarity.

Response: Materials and Methods" section as suggested, ensuring comprehensive and detailed descriptions are provided

Results

Strengths: The results section presents histological and biochemical findings comprehensively, supported by appropriate figures and tables. There is a clear comparison between different experimental groups, aiding in the interpretation of the data.

Weaknesses: The section is dense and may be challenging for readers to follow. Some interpretations are embedded in the results, which should be reserved for the discussion section.

Suggestions for Improvement:

  • Use subheadings to organize the results, making the section more reader-friendly.

  • Move interpretative comments to the discussion section to maintain a clear distinction between findings and their implications.

Discussion

Strengths:The discussion thoroughly interprets the findings, linking them to the study's objectives and existing literature. The potential mechanisms underlying the observed effects of Sevoflurane and Fullerenol C60 are effectively discussed.

Weaknesses: The discussion is somewhat repetitive, reiterating points already made in the results section. There is limited discussion on the study's limitations and future research directions.

Suggestions for Improvement:

  • Avoid repetition by summarizing key points more succinctly.
  • Include a section on the study's limitations and propose specific areas for future research.

Response: Discussion has been rewritten

Response: Limitations of the study has been added.

Conclusion

Strengths: The conclusion succinctly summarizes the main findings and their significance.

It highlights the potential clinical applications of the findings.

Weaknesses: The conclusion does not sufficiently address the broader implications of the study.

There is no mention of any limitations or caveats.

Suggestions for Improvement:

  • Expand the conclusion to discuss the broader implications of the research in the context of diabetes treatment and ischemia-reperfusion injury management.
  • Mention any study limitations and suggest future research directions to provide a balanced view.

Response: The conclusion has been rewritten to clearly summarize the findings of the current study and suggest directions for future research.

Best regards